# Spectral Analysis of the QT Interval Increases the Prediction Accuracy of Clinical Variables in Brugada Syndrome

**DOI:** 10.3390/jcm8101629

**Published:** 2019-10-04

**Authors:** Daniel García-Iglesias, Francisco Javier de Cos, Francisco Javier Romero, Srujana Polana, José Manuel Rubín, Diego Pérez, Julián Reguero, Jesús María de la Hera, Pablo Avanzas, Juan Gómez, Eliecer Coto, César Morís, David Calvo

**Affiliations:** 1Cadiology Department, Hospital Universitario Central de Asturias, 33012 Oviedo, Spain; daniel.garciai@sespa.es (D.G.-I.); jmrl100@gmail.com (J.M.R.); dpdcardio@gmail.com (D.P.); josejucasa@yahoo.es (J.R.); jesusdelahera@gmail.com (J.M.d.l.H.); avanzas@secardiologia.es (P.A.); cesarmoris@gmail.com (C.M.); 2Instituto de Investigación Sanitaria del Principado de Asturias, 33012 Oviedo, Spain; uo167835@uniovi.es (J.G.); eliecer.coto@sespa.princast.es (E.C.); 3Grupo para la Modelización Matemática Avanzada (MOMA), Universidad de Oviedo, 33003 Oviedo, Spain; fjcos@uniovi.es; 4Department of Medicine, Hospital Civil de Guadalajara Fray Antonio Alcalde, Guadalajara 44100, Mexico; romero_fco@outlook.es; 5Jawaharla Nehru Medical College, Belgaum 590001, Kartaka, India; janapolana@yahoo.com; 6Department of Molecular Genetics, Hospital Universitario Central de Asturias, 33012 Oviedo, Spain; 7Universidad Católica de Murcia, 30202 Murcia, Spain

**Keywords:** Brugada syndrome, spectral analysis, diagnosis, sudden cardiac death, prognosis

## Abstract

(1) Background: The clinical management of Brugada Syndrome (BrS) remains suboptimal. (2) Objective: To explore the role of standard electrocardiogram (ECG) spectral analysis in diagnosis and risk stratification. (3) Methods: We analyzed 337 patients—43 with a spontaneous type I ECG pattern (Spont-BrS), 112 drug induced (Induct-BrS), and 182 with a negative response to the drug challenge (negative responders (NR)). ECGs were processed using the wavelet transform (high frequency: 85 to 130 Hz). (4) Results: The power of the high-frequency content in the ST segment (Total ST Power; nV^2^Hz^−1^10^3^) was higher in BrS compared with NR patients (Spont-BrS: 28.126 (7.274–48.978) vs. Induc-BrS: 26.635 (15.846–37.424) vs. NR: 11.13 (8.917–13.343); *p* = 0.002). No differences were observed between ECG patterns in BrS patients. However, the Total ST Power of the type II or III ECG in NR patients was lower than in the same ECG patterns recorded from BrS patients (BrS: 31.07 (16.856–45.283); vs. NR: 10.8 (7.248–14.352) nV^2^Hz^−1^10^3^; *p* = 0.007). The Total ST Power, age, and family history of BrS were independent predictors of positive responses to drug testing. Comparing models with versus those without Total ST Power, the area under the received operator curve (ROC) curve increased (with 0.607 vs. without 0.528, *p* = 0.001). Only syncope was associated with an increased risk (follow-up 55.8 ± 39.35 months). However, the area under the ROC curve increased significantly when the Total ST Power was included as a covariate (with 0.784 vs. without 0.715, *p* = 0.04). (5) Conclusions: The analysis of the high-frequency content of ECG signals increases the predictive capability of clinical variables in BrS patients.

## 1. Introduction

Brugada syndrome (BrS) is an inherited disease with an increased risk of Sudden Cardiac Death (SCD) in apparently healthy individuals [1]. The diagnosis relies on the demonstration of a type I electrocardiogram (ECG) pattern, either occurring spontaneously or induced by the infusion of sodium channel blockers. However, the latter is questioned because of the suboptimal sensitivity of drug testing, which may negatively affect the prognosis in patients with false-negative responses [2]. Similarly, the intermittence of ECG patterns introduces a challenge to risk stratification and explains the conflicting results along different studies [3]. In fact, a significant portion of patients are reclassified with time and with an increasing number of ECG explorations [2].

Those limitations inherent to the visual inspection of ECG tracings might be overcome by quantitative analysis of the ECG signals. For that purpose, we previously demonstrated that the spectral decomposition of ECG signals with the wavelet transform of the QRS complexes allows for appropriate characterization of the high-frequency content, which exert a differential behavior between healthy individuals and patients affected by severe cardiac arrhythmias leading to SCD [4]. In the present work, we analyze an extensive cohort of patients with BrS and provide evidence of the potential utility of the spectral decomposition of ECG signals in improving the performance of diagnostic maneuvers and the accuracy of risk assessment beyond other variables commonly used in the clinic.

## 2. Methods

### 2.1. Population and Recording Protocol

From April 2005 to July 2018, data were collected from 337 patients with suspicious or confirmed BrS who were referred to our arrhythmia unit for diagnostic or therapeutic purposes (Appendix A). Patients were managed according to accepted recommendations at the time of evaluation [5] and classified as spontaneous BrS patients (Spont-BrS; patients displaying a spontaneous type I ECG pattern at the time of diagnosis), drug-induced BrS patients (Induc-BrS; patients displaying a type I ECG pattern during provocative testing with sodium blockers) and negative responder patients (NR; patients with suspicious BrS and a negative response to the provocative testing with sodium blockers).

Clinical baseline variables were obtained at the outpatient clinic. Patients displaying a spontaneous type I ECG were confirmed as having BrS, underwent risk stratification, and were referred for standard digital 12-lead ECG acquisition (see below). Patients with suspected BrS were referred for provocative testing with sodium blockers. According to recommendations, intravenous flecainide was continuously infused at a rate of 2.0 mg/kg body weight over 10 min (maximum dosage, 150 mg) [6]. Ajmaline was continuously infused at a rate of 1 mg/kg body weight over 10 min (maximum dosage, 50 mg). Before drug infusion, we checked for the absence of a type I ECG, both at the standard precordial position (V1 and V2 at the fourth intercostal space) and the high precordial position (V1 and V2 at the second intercostal space). At the end of the provocative testing, we explored the high precordial position for better sensitivity. ECG tracings were analyzed by two independent cardiac electrophysiologists and classified by consensus according to published recommendations as type I, II, or III [5]. The provocative testing was considered to display a positive response if the patient exhibited a type I ECG at any time during the protocol. The patients were retrospectively reviewed, and this study protocol was approved by the Ethics Committee. All patients gave informed consent.

### 2.2. Signal Processing

Standard ECGs (12 leads) were digitally used to extract the QT complexes (see Appendix A for details) [4]. The time–frequency data of each QT complex were collected using the Wavelet transform (see Appendix A for details). In accordance with previous reports, high-frequency content was defined as being within the range of 85 to 130 Hz [4]. Calculations were performed with R software (http://www.r-project.org) [7].

To analyze the distribution of the high-frequency content, we computed the cumulative power contained at each time epoch of the QT interval (Figure 1). From the obtained distribution, we defined (i) the Peak Power as the highest cumulative power of the high-frequency content, (ii) the Total Power as the area under the curve of the whole power function, (iii) the Total QRS Power as the area under the curve of the power function along the QRS interval, (iv) the Total ST Power as the area under the curve of the power function along the ST–T wave interval, and (v) the QRS to ST Total Power ratio as the ratio between the Total QRS Power and Total ST-Power.

### 2.3. Definitions

The terms sudden cardiac arrest (SCA) and sudden cardiac death (SCD) have been defined previously in the literature [8]. Symptomatic patients were defined according to the presence of any type of syncope [2,9,10]. The end point of this study was the occurrence of SCA, SCD, or appropriate therapy using an implantable defibrillator (ICD) to treat life-threatening ventricular arrhythmias during the follow-up period.

### 2.4. Follow-Up

Spont-BrS and Induc-BrS patients had an annual follow-up at the outpatient clinic. Risk stratification was performed according to current clinical standard recommendations, taking patient preferences into consideration. An electrophysiological study was also performed according to the state-of-the-art methods at the time. The induction of sustained ventricular fibrillation was followed by preventive ICD implantation. Alternatively, an ICD was recommended for high-risk patients, including SCA survivors and symptomatic patients. In contrast, patients displaying a negative response were not stratified according to the BrS standards. Every patient was also directly interviewed in the outpatient clinic at the time of this study, and data regarding the clinical profile were re-checked if necessary.

### 2.5. Statistical Analysis

Categorical variables are reported as numbers and percentages. Continuous variables are reported as means (±standard deviation [SD] or 95% Confidence Intervals [CI95%]). The chi-square test and the Student *t* test (paired or unpaired as appropriate) were used for univariate analysis to contrast different variables. For multilevel univariate analysis, an ANOVA test was used. Logistic regression was used to contrast different variables as predictors of the responses to provocative testing and SCA/SCD/appropriate therapies from the ICD during follow-up. A received operator curve (ROC) was constructed in both cases to evaluate the diagnostic and prognostic accuracy of the multivariate analysis, comparing the different models with the deLong test. Analyses were performed using R software (http://www.r-project.org), and statistical significance was established at *p* < 0.05.

## 3. Results

### 3.1. Patients and Clinical Variables

The distribution of patients and clinical characteristics are summarized in Appendix A and Table 1, respectively. Overall, BrS patients were slightly older than NR patients. Most of the Spont-BrS patients displayed a type I ECG pattern at the time of the digital ECG recording. Digital ECG records, acquired at the beginning of provocative testing, exhibited some differences between Induct-BrS and NR patients. Thus, most of the patients with a negative response to the provocative testing exhibited a normal ECG pattern at baseline, whereas the most frequent ECG pattern at baseline in the Induct-BrS cohort was the type II ECG pattern. The presence of syncope was equally distributed between groups. However, cardiac syncope and SCA were more frequent in BrS patients. An ICD was implanted in 22 Spont-BrS patients (51.16%) and in 23 Induct-BrS patients (20.54%), mainly because of sustained ventricular fibrillation induction in the electrophysiological study (11 patients; 24.44%) or previous cardiogenic syncope (13 patients; 28.89%)].

### 3.2. The High-Frequency Content along the QT Interval

The distribution of the high-frequency content along the QT interval was different between BrS patients and NR patients (Figure 2 and Table 2). Overall, the Total Power and the Total ST Power were significantly higher in BrS patients (either Spont-BrS or Induct-BrS) compared with NR patients. However, those differences were mainly determined by the differences observed in the right precordial leads (V1 and V2; See Table 2), while comparisons between other precordial leads displayed non-significant differences (V3 to V6; Total Power: Spont-BrS 27.932 (13.393–42.471) vs. Induc-BrS 42.991 (14.673–71.31) vs. NR patients 26.686 (22.626–30.747) 10^3^nV^2^Hz^−1^, *p* = 0.483; Total ST Power: Spont-BrS 12.821 (2.356–23.286) vs. Induc-BrS 12.735 (7.447–18.023) vs. NR patients 7.815 (6.139–9.49) 10^3^nV^2^Hz^−1^; *p* = 0.062).

When BrS patients were analyzed according to the time-domain description of the ECG records, we found no statistically significant differences between ECG patterns with regard to their high-frequency content (Appendix A). However, we observed significant differences in the Total ST Power contained in type II or III ECG patterns (combined) when comparing NR with BrS patients (Table 3). Such differences were not found when we compared normal ECG patterns from BrS patients with those from NR patients (see Appendix A for detailed description). An independent analysis of type II and III ECG patterns is presented in Appendix A. In summary, significant differences regarding the Total ST Power were identified when comparing Brugada patients and NR displaying a type II ECG pattern. Those differences were not observed when analyzing individuals displaying a type III ECG pattern. However, the number of patients available for analysis in that category was low, and therefore, the results were probably affected by a lack of statistical power.

### 3.3. Drug Challenge and the High-Frequency Content

Overall, 294 patients were admitted for drug challenge testing, and digitalized ECG records were obtained (baseline ECG records). We analyzed the diagnostic yield of the high-frequency content to predict positive responses to the test. In summary, 182 patients were classified as NR and 112 were classified as Induct-BrS. Univariate analysis demonstrated the Total Power and the Total ST Power at the right precordial leads, along with age, male gender, and family history of SCD or BrS, as variables with significant associations with the final drug testing results (Table 4). In the Multivariate Analysis, the Total ST Power, age, and family history of BrS were also found to be independent predictors of the final drug testing results (Table 4). Compared with a simplified model including age and family history of BrS, a completed model including the Total ST Power displayed an increased diagnostic yield. The inclusion of the Total ST Power significantly increased the ROC area under the curve compared with the simplified model (AUC completed model 0.607 vs. simplified model 0.528, *p* = 0.001; Figure 3A).

In a subset of 211 patients, digitalized ECG data were also collected after Flecainide (n = 168) or Ajmaline (n = 43) infusion. In that cohort, 61 patients displayed a type I ECG pattern after drug testing and were subsequently classified as Induct-BrS patients. Overall, drug infusion attenuated the high-frequency content along the QT interval in all individuals (Figure 4 and Appendix A). As displayed in Figure 4, no significant differences were observed in the rate of attenuation when comparing Induct-BrS patients with NR patients. In addition, Flecainide and Ajmaline attenuated the high-frequency content in a similar way (see Appendix A for details).

### 3.4. Prediction of Clinical Events During Follow-Up in Patients with Brugada Syndrome

The mean follow-up period was 55.8 ± 39.4 months (no patient was lost to follow-up). Overall, 14 patients (five Spont-BrS and nine Induct-BrS) had SCA or received appropriate ICD therapies because of ventricular fibrillation (9.03%). BrS patients with clinical events expressed a non-significant increase in the Total ST Power (51.18 [4.37–97.99] vs. 24.32 [14.86–33.79], *p* = 0.248) and in the Total ST Power along the right precordial leads (113.52 [–14.5 to 241.54] vs. 49.1 [25.17–73.04], *p* = 0.307). It translates into a significant reduction in the QRS to ST Total Power ratio compared with BrS patients without clinical events (Appendix A). In the univariate and multivariate analyses, cardiac syncope was the unique variable associated with an increased risk of clinical events (Table 4). However, the inclusion of the Total ST Power contained in right precordial leads in addition to cardiac syncope resulted in the increased predictive capability of the model. As shown in Figure 3B, the completed model, including syncope and Total ST Power, increased the ROC area under the curve significantly compared with the model with syncope alone (completed model AUC 0.784 vs. only syncope model AUC 0.715, *p* = 0.04). Comparisons between BrS patients displayed that those with clinical events expressed a significant reduction in the QRS to ST Total Power ratio compared with asymptomatic BrS patients (Appendix A).

## 4. Discussion

The results of our study show that the analysis of the high-frequency content of surface ECG signals adds diagnostic and prognostic information in BrS patients, as it helps to increase the predictive capability of clinical variables. We demonstrated that the high-frequency content exerts differential behaviors between BrS patients and controls, which is, to some extent, independent of the time domain classification of ECG patterns. Moreover, despite this differential behavior, the clinical significance shown in the ROC analysis for this parameter seems low compared with what was seen for the event prediction analysis. Because of that and although their role in BrS pathophysiology was not demonstrated in our work, the improvement in predictive capabilities adds more evidence in favor of the previously reported link between the high-frequency content and the risk for severe cardiac arrhythmias [4].

We are aware that translation to the clinic is far from being done; however, with the present work, we have paved the way for new quantitative measurements on ECG signals with the potential to improve the clinical management of BrS patients.

### 4.1. The Plausible Link between the High-Frequency Content and the Arrhythmogenic Substrate

Recent studies in BrS patients demonstrated that the arrhythmogenic substrate is confined to the epicardial layer of the right ventricle out-flow tract and free wall [11,12]. The electrograms recorded from the substrate characteristically displayed abnormal high-frequency potentials, expanding the length of the QRS interval and occupying positions at the ST segments. The abolition of such abnormal potentials has been proposed as a promising effective therapy that is able to reverse the type I ECG pattern and control arrhythmia recurrence. If the previous assumption is true, signal processing tools able to quantify the high-frequency content in the QT complexes might non-invasively characterize the arrhythmogenic substrate of BrS patients.

The signal average is the classical method applied to time domain records and has been postulated to have potential utility in the risk stratification of BrS patients [13,14,15,16,17,18]. However, the signal-averaged ECG is highly dependent on noise and requires long time records, which makes it tedious to use and has never previously helped to provide clear recommendations for patient management. In contrast, we and others previously demonstrated that the continuous wavelet transform may provide efficient analysis of the QRS signal, enabling the identification of late potentials by their subrogate in the frequency domain: the high-frequency content [4,19]. We hypothesized that the high-frequency content of the QT interval may correlate with the high-frequency electrograms founded as the arrhythmogenic substrate in BrS patients. The latter remains speculative but is strongly supported by the data displayed in our work, which provides an incentive for future research.

### 4.2. The High-Frequency Content and Patient Prognosis in BrS

Symptoms are major clinical determinants of prognosis in BrS patients, leading to conservative approaches when considering ICD implantation in asymptomatic patients. Despite the possibility of a selection bias of survivors that precludes accurate estimations of the real incidence of SCA/SCD in the general population with BrS [20], most clinical series have demonstrated good prognosis of asymptomatic patients under close follow-up and management of lifestyle, avoidance of drugs with potential adverse effects, and prompt treatment of fever [21]. Under such conditions, the annual incidence of SCA/ICD therapies varies within the range of 0.5% to 1% [22]. However, more than 50% of SCA episodes may occur in previously asymptomatic patients [23], and the cumulative risk has been demonstrated as stable over time [24], which might lead the incidence of arrhythmic events to rise by up to 10% in the next 10 years. This is unacceptable from a clinical point of view and highlights the necessity for clinical improvements in risk stratification in order to prevent rare but devastating events.

Several ECG features may help in risk stratification including fragmentation of the QRS, association with early repolarization syndrome, increased Tpeak–Tend intervals, quantitative measurements on the terminal R wave in lead V1, or the extension of the PR interval [22]. These measurements are widely available in the clinic, as they can be easily performed on a standard ECG. However, the implementation of the automatic quantification of ECG properties might help to overcome subjective interpretation on the ECG tracings and errors occurring when performing hand-made measurements. As presented in our work, BrS patients behave with an increased high-frequency content along the QT interval compared with controls. This difference is highlighted in patients with type II or type III Brugada patterns, which are more challenging ECG presentations. In fact, the presence of increased high-frequency content is an independent predictor of BrS during the drug challenge test, which significantly increases the diagnosis accuracy of other described variables (i.e., age and family history of BrS) and increases the accuracy of syncope as a predictor of events in BrS patients.

In conclusion, our study shows that the high-frequency content of the QT complexes exerts differential behavior in BrS patients that may be linked to the arrhythmogenic substrate and provides additional information for the time domain classification of ECG patterns. Further investigation is needed to establish the roles of these factors as independent predictors of fatal events in the global population with BrS.

## 5. Limitations

Data regarding the clinical profiles and the characteristics of episodes of syncope were re-checked by direct interviews with the subjects of interest at the time of this study. Thus, we cannot be sure that the patients’ memories regarding the conditions of syncope were accurate, which might be an important limitation when concluding the nature of syncope.

This study is observational and retrospective; thus, potential biases may arise because of missing data or inaccurate information collection. A second evaluation with other cohorts would be of interest for external validation. In addition, the number of patients included for analysis was low when attempting the analysis of subgroups (i.e., patients displaying the type III ECG pattern). The latter may have affected appropriate conclusions being reached.

## Figures and Tables

**Figure 1 jcm-08-01629-f001:**
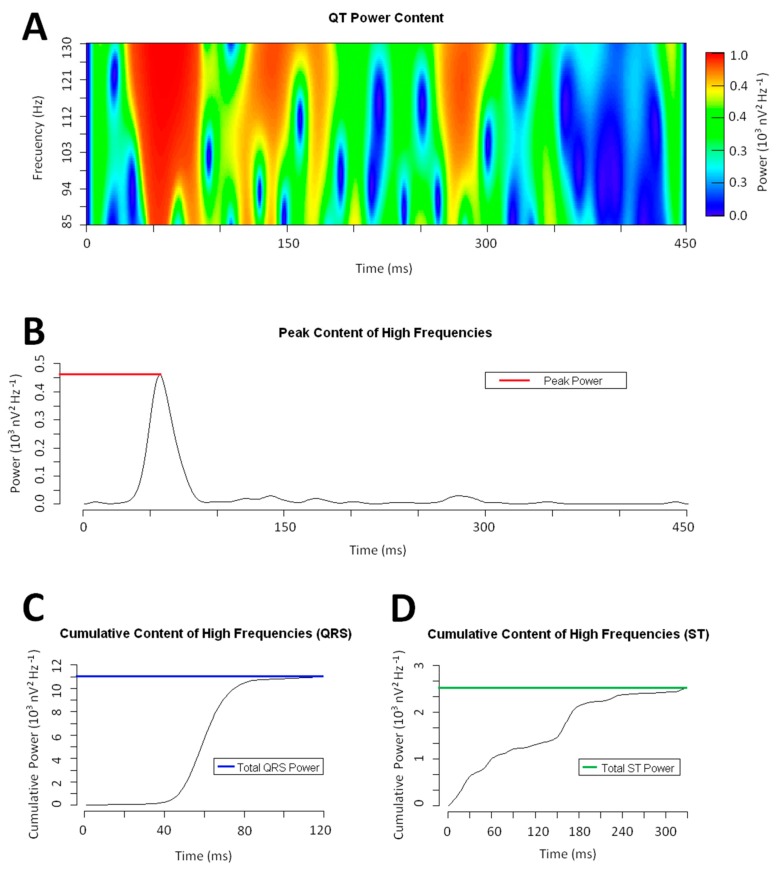
Example of a wavelet continuous transform on a QRS complex (frequency range: 85–130 Hz). **Panel A**: Power spectrum of the QRS complex. **Panel B**: Total high-frequency content at each time epoch. The brown dotted line marks the Peak Power. **Panel C & D**: Cumulative power of the high-frequency content along the QRS and ST interval. The colored dotted lines mark the Total QRS and ST Power respectively.

**Figure 2 jcm-08-01629-f002:**
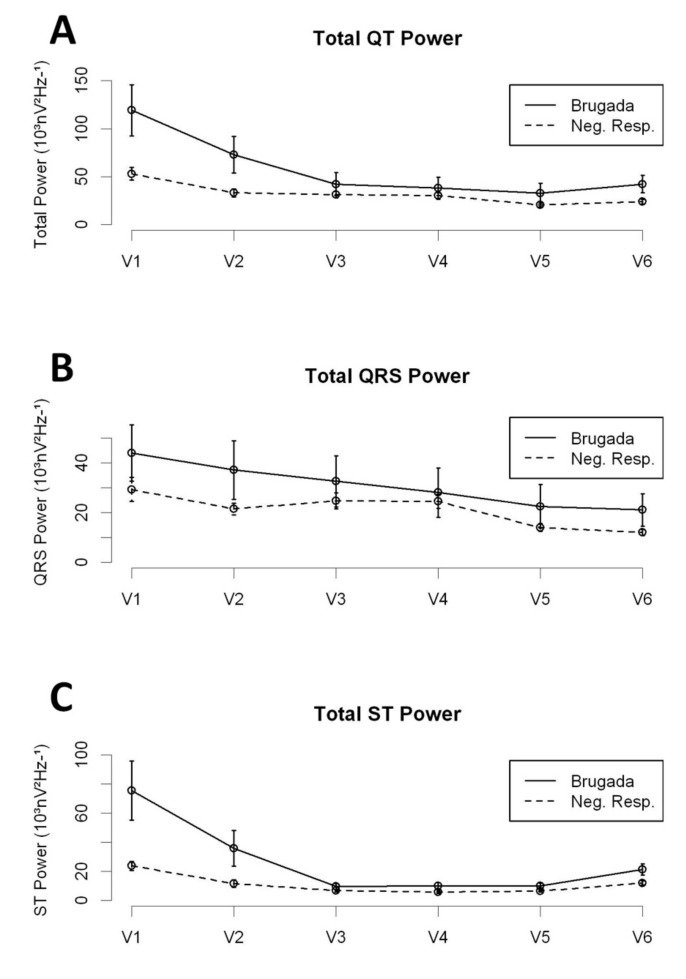
High-frequency content along precordial leads; Comparison between BrS patients and NR. **Panel A**: Total Power in the QT interval. **Panel B**: Total Power in the QRS interval. **Panel C**: Total Power in the ST interval.

**Figure 3 jcm-08-01629-f003:**
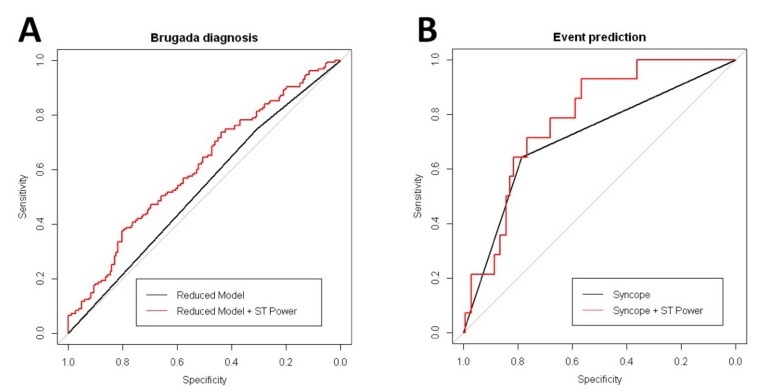
Comparative received operator curve (ROC) curve analysis from multivariate models. **Panel A**: ROC curve for BrS diagnosis during the drug testing. **Panel B**: ROC curve for arrhythmic event prediction.

**Figure 4 jcm-08-01629-f004:**
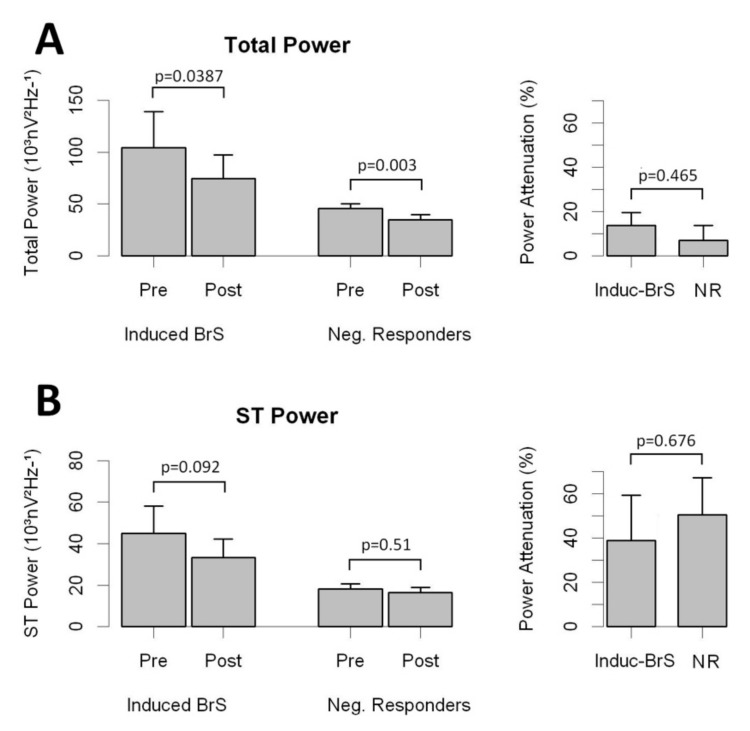
Effects of drug infusion on the high-frequency content of the QT interval for right precordial leads. **Panel A**: Total Power and attenuation of Total Power in Brugada patients and NR. **Panel B**: ST Power and attenuation of ST Power in Brugada patients and NR.

**Table 1 jcm-08-01629-t001:** Comparison of clinical variables between groups.

	Spont-BrS (N = 43)	Induc-BrS (N = 112)	NR Patients (N = 182)	*p* Value
***Clinical features***				
Age (years)	44.05 (12.3)	43.61 (14.51)	38.64 (14.98)	**0.004**
Male gender (%)	30 (90.7)	70 (62.5)	137 (75.28)	**0.001**
Family history of SCD at age <45 years (%)	18 (41.86)	68 (60.71)	73 (40.11)	**0.002**
Syncope (%)	11 (25.58)	28 (25)	56 (30.77)	0.521
Cardiac syncope (%)	7 (16.28)	12 (10.71)	5 (2.75)	**0.002**
SCA (%)	5 (11.63)	9 (8.04)	1 (0.549)	**0.001**
Smoker (%)	12 (27.9)	29 (25.89)	47 (25.82)	0.96
Hypertension (%)	7 (16.28)	18 (16.07)	21 (11.54)	0.473
Diabetes mellitus (%)	1 (2.33)	4 (3.57)	3 (1.65)	0.575
Dyslipidemia (%)	8 (18.61)	22 (19.64)	14 (7.69)	**0.007**
Cardiomyopathy (%) ^†^	3 (6.98)	3 (2.68)	9 (4.95)	0.455
Cardiovascular drugs (%) ^‡^	11 (25.58)	18 (16.07)	23 (12.64)	0.104
PES Test performed	26 (60.47)	37 (33.04)	3 (1.65)	**<0.001**
Positive PES	8 (18.6)	4 (3.57)	0 (0)	**<0.001**
ICD implanted	22 (51.16)	23 (20.54)	2 (1.1)	**<0.001**
***ECG pattern at the time of the digital record***				
BrS type I (%)	38 (88.37)	0	0	**<0.001**
BrS type II (%)	3 (6.98)	59 (52.68)	36 (19.78)	**<0.001**
BrS type III (%)	0	22 (19.64)	39 (21.43)	**0.004**
BrS type II–III (%)	3 (6.98)	81 (72.62)	75 (41.21)	**<0.001**
Normal (%)	0	25 (22.32)	75 (41.21	**<0.001**

† All the cases displayed discrete left ventricle hypertrophy due to hypertension. ‡ All the cases on anti-hypertensive and/or lipid-lowering drugs. BrS: Brugada syndrome; SCA: sudden cardiac arrest; SCD: sudden cardiac death; Spont-BRS: spontaneous BrS patients; Induc-BRS: drug-induced BrS patients; NR: negative responder patients; PES: programmed electrical stimulation.

**Table 2 jcm-08-01629-t002:** Comparative analysis of the high-frequency content between different clinical conditions.

	Spont-BrS	Induct-BrS	NR Patients	*p* Value
***All precordial leads***				
Peak Power	0.734 (0.616–0.852)	1.439 (0.916–1.962)	0.871 (0.786–0.956)	0.677
Total Power	46.693 (34.811–58.575)	62.188 (46.143–78.233)	32.161 (29.752–34.57)	0.095
Total QRS Power	18.567 (15.884–21.25)	35.553 (22.559–48.547)	21.031 (19.119–22.943)	0.623
Total ST Power	28.126 (17.793–38.459)	26.635 (21.19–32.08)	11.13 (10.009–12.251)	0.002
QRS to ST Total Power	5.256 (3.947–6.565)	5.762 (4.931–6.593)	9.724 (8.075–11.373)	0.045
***Right precordial leads***				
Peak Power	0.897 (0.74–1.054)	1.705 (1.127–2.283)	0.917 (0.801–1.033)	0.468
Total Power	84.216 (52.704–115.728)	100.581 (77.381–123.781)	43.111 (38.832–47.39)	0.017
Total QRS Power	25.48 (21.46–29.5)	46.147 (30.805–61.489)	25.35 (22.267–28.433)	0.451
Total ST Power	58.736 (30.649–86.823)	54.434 (40.921–67.947)	17.761 (15.586–19.936)	0.003
QRS to ST Total Power	4.142 (3.075–5.209)	4.06 (3.445–4.675)	6.023 (5.067–6.979)	0.133

Figures within brackets denote the 95% confidence interval (CI95%). Units for Peak Power, Total Power, Total QRS Power, and Total ST Power are expressed as 10^3^nV^2^Hz^−1^.

**Table 3 jcm-08-01629-t003:** Comparative analysis of the high-frequency content between different electrocardiogram (ECG) patterns and clinical conditions.

	ECG Type I	ECG Type II or III		
	BrS Patients	BrS Patients	NR Patients	*p*
***All precordial leads***				
Peak Power	0.629 (0.421–0.836)	1.518 (0.186–2.85)	1.07 (0.762–1.379)	0.517
Total Power	47.415 (20.269–74.561)	69.721 (28.191–111.251)	36.259 (27.264–45.253)	0.121
Total QRS Power	16.665 (11.358–21.972)	38.651 (5.058–72.244)	25.458 (18.461–32.455)	0.446
Total ST Power	30.75 (7.171–54.329)	31.07 (16.856–45.283)	10.8 (7.248–14.352)	0.007
QRS to ST Total Power	3.849 (2.131–5.566)	5.853 (3.926–7.779)	12.132 (6.002–18.262)	0.055
***Right precordial leads***				
Peak Power	0.886 (0.529–1.244)	1.948 (0.43–3.466)	1.209 (0.771–1.647)	0.355
Total Power	89.832 (17.724–161.941)	120.243 (59.55–180.936)	51.683 (35.198–68.167)	0.033
Total QRS Power	25.041 (15.957–34.126)	53.695 (13.362–94.027)	32.757 (20.998–44.516)	0.324
Total ST Power	64.791 (0.574–129.008)	66.549 (31.128–101.969)	18.926 (11.609–26.242)	0.01
QRS to ST Total Power	3.471 (1.768–5.173)	4.284 (2.783–5.785)	7.666 (3.581–11.751)	0.125

Figures within brackets denote the CI95%. Units for Peak Power, Total Power, Total QRS Power, and Total ST Power are expressed as 10^3^nV^2^Hz^−1^.

**Table 4 jcm-08-01629-t004:** Results of the Univariate and Multivariate analyses.

	Univariate		Multivariate	
	HR	*p*	HR	*p*
***Model for prediction of positive response to the drug challenge***				
Peak Power	3.251 (0.8–13.209)	0.099		
Total Power	1.054 (1.019–1.091)	0.003		
Total QRS Power	1.045 (0.991–1.102)	0.101		
Total ST Power	1.106 (1.043–1.174)	0.001	1.251 (1.082–1.447)	0.003
QRS to ST Total Power ratio	0.678 (0.407–1.13)	0.136		
Age	1.005 (1.002–1.009)	0.006	1.005 (1.001–1.008)	0.014
Male	0.865 (0.766–0.977)	0.02	0.925 (0.814–1.05)	0.225
Familiar History of SCD	1.215 (1.089–1.356)	0.001		
Familiar History of BrS	1.203 (1.066–1.358)	0.003	1.158 (1.019–1.317)	0.025
Syncope	0.936 (0.827–1.059)	0.289	0.914 (0.81–1.032)	0.146
***Model for prediction of arrhythmic events during follow-up***				
Peak Power	0.997 (0.414–2.398)	0.994		
Total Power	1.011 (0.991–1.031)	0.285		
Total QRS Power	0.999 (0.967–1.033)	0.967		
Total ST Power	1.025 (0.996–1.056)	0.096	1.041 (0.966–1.123)	0.291
QRS to ST Total Power ratio	0.536 (0.27–1.065)	0.075		
Age	1 (0.997–1.003)	0.905		
Spontaneous Type I Pattern	1.037 (0.936–1.148)	0.488	1.026 (0.923–1.141)	0.629
Male	1.036 (0.938–1.145)	0.482	1.041 (0.939–1.155)	0.441
Familiar History of SCD	0.955 (0.871–1.047)	0.322	0.951 (0.869–1.041)	0.278
Familiar History of BrS	0.928 (0.842–1.023)	0.133		
Syncope	1.206 (1.09–1.335)	<0.001	1.197 (1.079–1.329)	0.001
Positive PES	0.907 (0.765–1.075)	0.259	0.898 (0.756–1.067)	0.219
SCN5a Mutation	0.96 (0.86–1.073)	0.472	0.975 (0.873–1.089)	0.652

Numbers within brackets denote the CI95%.

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
