# Peer review of "Spectral Analysis of the QT Interval Increases the Prediction Accuracy of Clinical Variables in Brugada Syndrome"

_jcm, 2019, doi:10.3390/jcm8101629_

Round 1
Reviewer 1 Report
This paper introduces spectral analysis of the ECG into the evaluation of Brugada syndrome. It is a single centre retrospective study, which is fine as an exploratory analysis and it comes with a large number of subjects. The findings are of potential interest in this difficult area.A less subjective ECG analysis would be very attractive
However such a study comes of course with the usual caveats- data missing or inaccurate, not obtained for this work, potential bias etc, and the study lacks a second evaluation cohort. These significant limitations should be addressed in the conclusions.
In terms of the data I would like some clarification because it seems, if I am right, that the comparisons are made between Brugada 1 and then Brugada 2 and 3 lumped together. More drug responders were seen with those with type 2 so this would not seem to be legitimate. More of interest would be comparing all type 3s, and all type 2s in terms of their spectral analysis, if I am right then this would require a separate, additional analysis.
There are some challenges in understanding the English and I would respectfully suggest that it would help reviewers if the manuscript was re-written by a native English speaker.
Reviewer 2 Report
The purpose of this study was to investigate the potential utility of spectral decomposition of ECG signals in increasing the diagnostic accuracy and risk assessment in patients with Brugada syndrome . The endpoint was the occurrence of cardiac arrest , sudden death or appropriate shocks by implanted ICD.
-The study is interesting , well written and performed in a large series of Brugada syndrome patients .
-The most clinically intriguing finding of the study is that the total ST power ( one of the parameters obtained with spectral decomposition of the ECG signals ) improved the ability of clinical variables ( cardiogenic syncope ) in predicting the occurrence of sudden cardiac death in the follow-up . The authors however should clarify why they used sudden cardiac death as endpoint for this analysis rather than the composite endpoint they declared in the Methods .
- The total ST power improved significantly the area under the curve of the statistical model ( including age and family history of Brugada syndrome ) to predict the result of drug testing , but the diagnostic yield was low ( 0.60 vs 0.52 ) a little bit more than tossing a coin. The authors should downplay in the Discussion the importance of this finding.
- The sentence in the Methods “ Electrophysiologic study was also performed according the state of the art at the time , but due to sensitivity and specificity concerns , considering preferences of both patients and clinicians “ is obscure . Please rephrase it also , considering that the main reason for ICD implantation ,in this series , as it comes out in the Results , of patients was induction of VF at the electrophysiologic study .
Round 2
Reviewer 1 Report
Thankyou for the changes. I have no further comments.